# M³E: Continual Vision-and-Language Navigation via Mixture of Macro and Micro Experts

**Yongliang Jiang**[1]    **Huaidong Zhang**[1,*]   **Xuandi Luo**[1]    **Shengfeng He**[2]

[1]South China University of Technology, [2]Singapore Management University

`{ftyljiang,ftxuandi.luo}@mail.scut.edu.cn, huaidongz@scut.edu.cn,`
`shengfenghe@smu.edu.sg`

## ABSTRACT

Vision-and-Language Navigation (VLN) agents have shown strong capabilities in following natural language instructions. However, they often struggle to generalize across environments due to catastrophic forgetting, which limits their practical use in real-world settings where agents must continually adapt to new domains. We argue that overcoming forgetting across environments hinges on decoupling global scene reasoning from local perceptual alignment, allowing the agent to adapt to new domains while preserving specialized capabilities. To this end, we propose M³E, the *Mixture of Macro and Micro Experts*, an environment-aware hierarchical MoE framework for continual VLN. Our method introduces a dual-router architecture that separates navigation into two levels of reasoning. A macro-level, scene-aware router selects strategy experts based on global environmental features (e.g., office vs. residential), while a micro-level, instance-aware router activates perception experts based on local instruction-vision alignment for step-wise decision making. To preserve knowledge across domains, we adopt a dynamic momentum update strategy that identifies expert utility in new environments and selectively updates or freezes their parameters. We evaluate M³E in a domain-incremental setting on the R2R and REVERIE datasets, where agents learn across unseen scenes without revisiting prior data. Results show that our method consistently outperforms standard fine-tuning and existing continual learning baselines in both adaptability and knowledge retention, offering a parameter-efficient solution for building generalizable embodied agents. Our project page is available at `https://yongliangjiang.top/m3e`.

## 1 INTRODUCTION

Vision-and-Language Navigation (VLN) tasks require embodied agents to follow free-form natural language instructions to reach target locations in photorealistic indoor environments Anderson et al. (2018); Qi et al. (2020); Ku et al. (2020); Thomason et al. (2020); Zhu et al. (2021). This setting demands the tight integration of visual perception, language grounding, and sequential decision making under partial observability. Significant progress has been made in recent years. Early approaches focused on aligning panoramic visual inputs with language using multimodal encoders and imitation learning Hao et al. (2020); Majumdar et al. (2020), often leveraging large-scale vision-language pretraining Chen et al. (2020); Tan & Bansal (2019) and data augmentation to improve generalization Tan et al. (2019); Ma et al. (2019a). Beyond cross-modal alignment, subsequent work introduced memory modules Hong et al. (2021), backtracking mechanisms Ma et al. (2019b), and topological map construction Chen et al. (2021a) to support long-horizon reasoning. More recently, large language models (LLMs) have been incorporated to enhance world knowledge access and compositional reasoning, either through prompting or fine-tuning Pan et al. (2023); Zheng et al. (2023); Zhang et al. (2024).

Despite these advances, most VLN systems are trained on static datasets and exhibit poor generalization to unseen environments. Adapting to new domains often requires expensive retraining and can lead to catastrophic forgetting of previously learned skills. Continual learning in VLN remains

---

*Corresponding Author.

underexplored. Existing methods primarily rely on rehearsal buffers that replay past trajectories during training Jeong et al. (2024); Li et al. (2024), which increases storage and computational overhead and may raise privacy concerns. Furthermore, VLN presents unique challenges not encountered in classification tasks (e.g., CL-MoE Huai et al. (2025)), such as the need for sequential planning under partial observability and fine-grained instruction grounding.

We argue that overcoming forgetting in continual VLN requires separating high-level scene reasoning from low-level perceptual alignment. Scene-level understanding captures structural patterns that generalize across domains, such as layouts typical of offices or residential spaces. In contrast, token-level grounding provides fine-grained cues for local decision making. Conflating these two levels of reasoning can result in brittle policies and poor transfer across tasks.

To address this, we propose M$^3$E, a **M**ixture of **M**acro and **M**icro **E**xperts for continual vision-and-language navigation. M$^3$E is a replay-free continual learning framework based on an end-to-end trainable LLM agent. It introduces a hierarchical mixture-of-experts architecture with dual routing to separate global and local reasoning. The macro router converts the agent's cognitive map of visited and frontier nodes into a sparse graph, performs topology-aware propagation using a graph neural network, and attends over nodes using instruction embeddings to produce scene-level expert weights. In parallel, the micro router derives token-wise expert activations from the hidden states of the LLM. These two routing signals are fused to select experts in MoE-LoRA layers, enabling context-sensitive specialization while maintaining sparse and efficient computation.

To support continual learning without replay, M$^3$E introduces a dynamic momentum update strategy that consolidates knowledge across domains. For each layer, we track expert usage by accumulating the fused routing weights over all tokens in the current task, normalize to obtain per-expert contributions, and select the top-performing experts. These are updated with larger momentum, while less relevant experts are updated conservatively to preserve generalizable knowledge. This mechanism enables rapid adaptation to new environments while mitigating forgetting of prior domains.

We evaluate M$^3$E in a domain-incremental setting on the R2R and REVERIE benchmarks, where agents learn in disjoint scene sets without revisiting prior data. We report standard VLN metrics such as success rate (SR) and success weighted by path length (SPL), along with continual learning metrics including forward transfer (FWT) and backward transfer (BWT). M$^3$E consistently outperforms strong baselines, improving both generalization and retention across domains.

Our contributions are as follows:

- We define the vision-and-language navigation continual learning (VLNCL) setting, with a task protocol and evaluation metrics, and present the first replay-free mixture-of-experts framework for this setting.

- We introduce M$^3$E, a dual-router hierarchical architecture that separates global scene reasoning from local token-level grounding, enabling more precise and modular expert specialization.

- We propose a dynamic momentum update mechanism that consolidates task-critical experts using differentiated update rates, supporting fast adaptation without data replay.

- We conduct extensive experiments on R2R and REVERIE, showing that M$^3$E improves both navigation success and continual learning performance, demonstrating stronger generalization and reduced forgetting across environments.

## 2 RELATED WORK

### 2.1 VISION-AND-LANGUAGE NAVIGATION

**Cross-modal grounding.** Early VLN methods focused on learning joint representations of panoramic visual inputs and language instructions Hao et al. (2020); Majumdar et al. (2020); Guhur et al. (2021); Chen et al. (2021b), typically using multimodal transformers. Subsequent improvements leveraged large-scale vision-language pretraining Chen et al. (2020); Li et al. (2020); Su et al. (2019); Tan & Bansal (2019), data augmentation Tan et al. (2019); Li et al. (2022); Dou & Peng (2022); Li & Bansal (2023), and advanced training regimes such as imitation learning combined with reinforcement learning Wang et al. (2018); Ma et al. (2019a); Zhu et al. (2020). These strate-

gies enhanced the agent's ability to ground linguistic references in visual observations and improved generalization to unseen environments.

**Planning and memory.** Beyond alignment, effective navigation requires reasoning over long trajectories. Several approaches have addressed this through memory-based encoders Hong et al. (2021); Chen et al. (2021c), self-correction and backtracking mechanisms Ke et al. (2019); Ma et al. (2019b), and explicit topological or semantic mapping An et al. (2021); Chen et al. (2021a); Liu et al. (2023); Wang et al. (2023); Liu et al. (2024). External knowledge sources and semantic priors have also been incorporated to guide planning in unfamiliar environments Lin et al. (2022); Li et al. (2023).

**Language models for navigation.** Recent work has explored using large language models (LLMs) to inject world knowledge and enable more structured reasoning Pan et al. (2023); Zhou et al. (2024; 2025); Qiao et al. (2025); Chen et al. (2024); Lin et al. (2024); Long et al. (2024); Zhang et al. (2024); Zheng et al. (2023). Some approaches treat the LLM as a frozen API, while others pursue end-to-end training of multimodal LLM-based agents Song et al. (2025); Mu et al. (2023); Zhang et al. (2024), aiming to unify language understanding, perception, and action within a single policy. These models demonstrate promising generalization, but still struggle to adapt continuously across diverse domains.

Despite these advances, most VLN agents are trained on fixed datasets and perform poorly when deployed in novel environments. Adapting to new domains often requires full retraining, which is computationally expensive and risks catastrophic forgetting of prior knowledge. These limitations motivate the development of continual learning (CL) capabilities for VLN agents.

## 2.2 CONTINUAL LEARNING FOR EMBODIED AGENTS

A critical capability for embodied agents is to learn incrementally across domains while preserving prior knowledge, a goal that CL explicitly targets. In the context of VLN, however, research in CL remains limited. Most existing approaches rely on *rehearsal-based* methods Jeong et al. (2024); Li et al. (2024), where past trajectories are stored and periodically replayed to mitigate forgetting. For example, PerpR/ESR Jeong et al. (2024) maintains a buffer of navigation episodes and interleaves them during training on new environments. While effective, these methods suffer from scalability and privacy concerns. Maintaining large buffers imposes storage and compute overhead, and replaying raw trajectory data raises concerns in applications where privacy or memory constraints are critical. To address these issues, recent work in related multimodal domains has explored *replay-free* solutions. For instance, CL-MoE Huai et al. (2025) proposes a Mixture-of-Experts (MoE) framework for continual visual question answering, where task-specific experts enable selective knowledge retention. However, this line of work has not yet been extended to VLN, which poses additional challenges due to its sequential, partially observable nature and the need for consistent spatial reasoning over time.

Our work bridges this gap by introducing a replay-free MoE-based framework specifically tailored for continual VLN. In contrast to prior CL approaches in vision or language tasks, we explicitly account for the hierarchical structure of navigation: separating global scene reasoning from local token-level grounding. This separation allows for targeted expert specialization, improves transferability, and supports continual adaptation without relying on trajectory replay.

## 3 PROBLEM FORMULATION

### 3.1 VISION-AND-LANGUAGE NAVIGATION CONTINUAL LEARNING (VLNCL)

A vision-and-language navigation (VLN) task is defined on an indoor environment represented as a navigation graph $\mathcal{G} = (\mathcal{V}, \mathcal{E})$, where nodes $\mathcal{V}$ are viewpoints and edges $\mathcal{E}$ denote navigable connections. Given a natural-language instruction $\mathbf{I} = (w_1, \ldots, w_L)$, an agent starts at node $v_0$ and sequentially observes panoramic views $o_t$ and selects actions $a_t \in \mathcal{A}(v_t)$ according to a policy $\pi_\theta(a_t \mid s_t, \mathbf{I})$, where $s_t$ denotes the agent's state at step $t$. Executing the policy produces a trajectory $\tau = (s_0, a_0, \ldots, s_T)$. Training generally seeks to minimize the expected cumulative loss:

$$\min_\theta \ \mathbb{E}_{(\mathcal{G}, \mathbf{I})} \ \mathbb{E}_{\tau \sim \pi_\theta} \Big[ \sum_{t=0}^{T} \mathcal{L}(s_t, a_t) \Big], \tag{1}$$

where $\mathcal{L}$ can be instantiated as imitation learning, reinforcement learning, or hybrid objectives.

In continual VLN, the agent faces a stream of tasks $\{D_t\}_{t=1}^T$, where each $D_t$ contains navigation graphs and language instructions from a (possibly) new domain. At stage $t$, the agent starts from consolidated parameters $\Theta_{t-1}$, adapts them on $D_t$ to obtain task-specific parameters $\Phi_t$, and then fuses them into new consolidated parameters $\Theta_t$:

$$\Phi_t \;=\; \arg\min_\theta \; \mathbb{E}_{(\mathcal{G},\mathbf{I})\sim D_t}\, \mathbb{E}_{\tau\sim\pi_\theta}\Big[\sum_k \mathcal{L}(s_k, a_k)\Big], \qquad \Theta_t \;=\; \mathcal{C}(\Theta_{t-1},\, \Phi_t), \qquad (2)$$

where $\mathcal{C}(\cdot,\cdot)$ is a consolidation operator (instantiated by our momentum update in Sec. 4.2). Unless otherwise noted, past-task data are not revisited (no rehearsal), making catastrophic forgetting a central challenge.

## 3.2 LLM-based Navigation Agent

Our agent follows the paradigm of end-to-end *trainable* LLM-based navigation, in the spirit of generalist models such as NaviLLM Zheng et al. (2024), EmbodiedGPT Mu et al. (2023), and NaVid Zhang et al. (2024). Unlike approaches that query frozen APIs, we directly fine-tune a multimodal LLM backbone on VLN. The system has two components: a *scene encoder* and a *language model*.

**Scene Encoder.** At each step the environment provides a discretized panoramic observation (e.g., 36 views). A pretrained Vision Transformer (ViT) first extracts per-view features, which are then fused by a lightweight multi-view encoder (Transformer or attentive pooling) to contextualize spatial relations across views and yield viewpoint-level scene embeddings. These embeddings update the cognitive map (visited and frontier nodes) used by downstream routing (Sec. 4.1).

**LLM Backbone and Action Generation.** We use a decoder-only Transformer (e.g., a 7B LLM) as the policy core. The instruction, navigation history, and scene embeddings are serialized into a unified input schema and fed to the LLM; the hidden state at the action query position is passed to a small action head to score reachable candidate viewpoints, defining a single-step autoregressive decision. The process repeats until STOP is emitted or the step budget is exhausted.

**Modification for M³E.** Our key architectural innovation is the replacement of the standard Feed-Forward Network (FFN) layers within the LLM backbone with our proposed M³E layers, as detailed in Sec. 4. This modification allows the agent to dynamically route computations through specialized experts based on both high-level strategic needs and low-level semantic context, while supporting continual learning across domains.

## 4 Method

We propose M³E, a framework that equips large language model-based VLN agents with continual learning capabilities. It consists of two main components. The Macro–Micro MoE (Sec. 4.1) uses a dual-router design: a Macro Router captures global, scene-level strategies, while a Micro Router handles local, token-level semantics. The Dynamic MoE Momentum Update (Sec. 4.2) reduces forgetting by identifying task-critical experts and updating them with differentiated momentum, allowing rapid adaptation while preserving transferable knowledge.

### 4.1 Mixture of Macro and Micro Experts

#### 4.1.1 Macro Router

The macro router $G^{\mathrm{ma}}$ is designed to capture global, structural regularities of the navigation environment and align them with the high-level task intent. Instead of naively aggregating visual information, we propose Topology-Aware, Task-Focused (TATF) Routing. This method uses a GNN to understand the structure of the agent's known world (the "Where"), then applies instruction-guided attention to focus on relevant areas (the "What").

**Step 1: Global Map Representation.** From the cognitive map available at the current step (including both nodes already visited and nodes discovered but not yet explored), we derive pairwise node

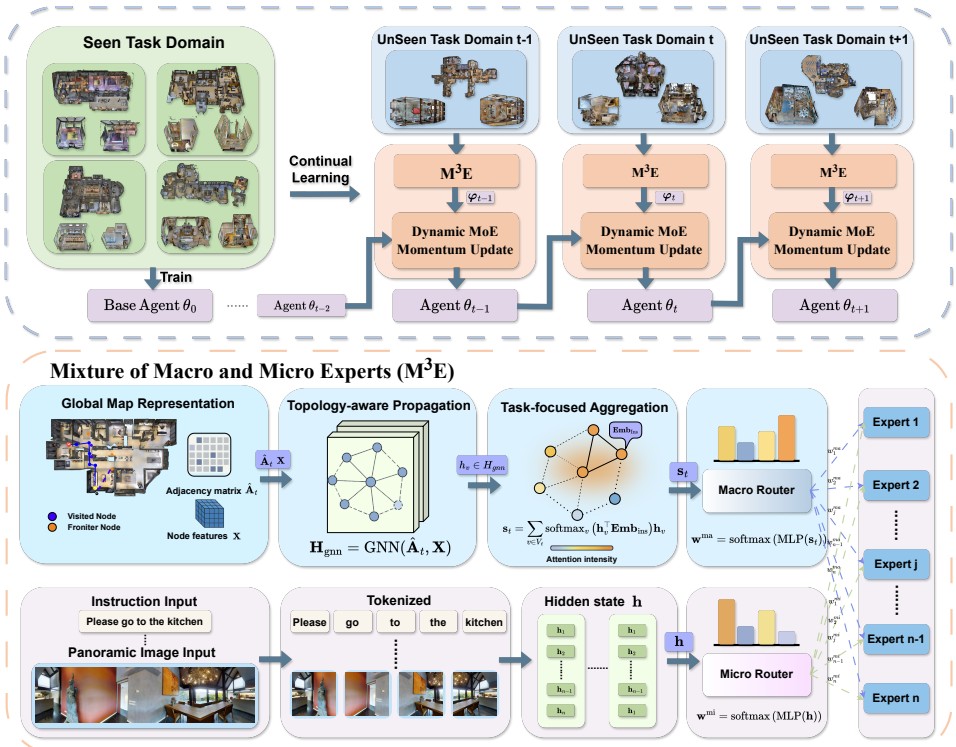

Figure 1: Overall architecture of M³E. The framework decouples macro-level scene reasoning and micro-level token grounding via a dual-router design. The Macro Router (blue) builds a task-aware scene representation using GNN-based propagation over a cognitive map. The Micro Router (purple) computes token-wise expert weights from hidden states. Both signals are fused to route experts in the MoE-LoRA layers for planning.

distances and threshold them to form a sparse adjacency $\hat{\mathbf{A}}_t$ encoding connectivity among $N$ nodes. Each node $v \in V_t$ is initialized with a feature vector $\mathbf{x}_v$ that integrates panoramic visual embeddings, spatial location, temporal step embeddings, and navigation-state indicators. These node features are collected into a matrix:

$$\mathbf{X} = \left[\mathbf{x}_v^\top\right]_{v \in V_t} \in \mathbb{R}^{N \times d}. \tag{3}$$

Importantly, the cognitive map is constructed *online* from the agent's exploration history, including visited and frontier nodes, rather than using any pre-defined global map.

**Step 2: Topology-aware Propagation** The node features $\mathbf{X}$ and adjacency matrix $\hat{\mathbf{A}}_t$ are fed into a Graph Neural Network (GNN). The GNN performs message passing to learn the spatial relationships between nodes, yielding topology-aware representations:

$$\mathbf{H}_{\mathrm{gnn}} = \mathrm{GNN}(\hat{\mathbf{A}}_t, \mathbf{X}) \in \mathbb{R}^{N \times d_h}. \tag{4}$$

**Step 3: Task-focused Aggregation.** Let $\mathbf{Emb}_{\mathrm{Ins}} \in \mathbb{R}^{d_h}$ be the embedding of the navigation instruction. We use $\mathbf{Emb}_{\mathrm{Ins}}$ as a query to compute attention scores on topology-aware node representations $\mathbf{H}_{\mathrm{gnn}}$ (where $\mathbf{h}_v$ is the representation for node $v$):

$$\alpha_v = \mathrm{softmax}_v\big(\mathbf{h}_v^\top \mathbf{Emb}_{\mathrm{Ins}}\big), \qquad \mathbf{s}_t = \sum_{v \in V_t} \alpha_v \, \mathbf{h}_v \in \mathbb{R}^{d_h}. \tag{5}$$

The resulting vector $\mathbf{s}_t$ is an aggregated scene representation that is both structurally informed and semantically focused on the current task.

**Step 4: Scene-level routing.** Finally, the aggregated scene vector $\mathbf{s}_t$ is passed through a routing head to produce the macro-level expert weights:

$$\mathbf{w}^{\mathrm{ma}} = G^{\mathrm{ma}}(\mathbf{s}_t) = \mathrm{Softmax}\big(\mathrm{MLP}(\mathbf{s}_t)\big) \in \mathbb{R}^n, \tag{6}$$

where $n$ is the number of experts. This routing mechanism provides the model with a richer, more actionable understanding of the scene, enabling more effective macro-level planning.

### 4.1.2 MICRO ROUTER

At the micro (token-level) granularity, we introduce a router $G^{\mathrm{mi}}$ that operates on each token embedding to produce a distribution over experts. Given the hidden state $\mathbf{h}$ of a token in the navigation step, the generic form follows the MoE gating paradigm:

$$\mathbf{w}^{\mathrm{mi}} = G^{\mathrm{mi}}(\mathbf{h}) = \mathrm{Softmax}\left(\mathrm{MLP}(\mathbf{h})\right) \in \mathbb{R}^n, \tag{7}$$

where $\mathbf{w}^{\mathrm{mi}} = [w_1^{\mathrm{mi}}, \ldots, w_n^{\mathrm{mi}}] \in \mathbb{R}^n$ denotes the token-wise routing weights over $n$ experts, and $w_i^{\mathrm{mi}}$ is the probability of choosing expert $E_i$. This micro router captures fine-grained semantics within the instruction stream (e.g., in "go to the kitchen," the verb token "go" is inclined to *action reasoning* experts, while the noun token "kitchen" prefers *object/scene understanding* experts), enabling context-sensitive expert specialization. We train $G^{\mathrm{mi}}$ on the task data $D_t$ so that it learns to select local experts conditioned on token-level representations.

### 4.1.3 DUAL-ROUTER FUSION

Finally, the macro router $G^{\mathrm{ma}}$ and the micro router $G^{\mathrm{mi}}$ are fused to produce the final expert routing weights. The macro weights $\mathbf{w}^{\mathrm{ma}}$ provide a global, scene-level prior that reflects the structural layout and high-level task intent, while the micro weights $\mathbf{w}^{\mathrm{mi}}$ offer token-level, fine-grained judgments that adapt to the current linguistic context. We combine them through a convex interpolation:

$$\mathbf{w} = \beta\,\mathbf{w}^{\mathrm{ma}} + (1 - \beta)\,\mathbf{w}^{\mathrm{mi}} \in \mathbb{R}^n, \tag{8}$$

where $\beta$ is a hyper-parameter that balances the importance of global priors and local adaptivity. This dual-router fusion equips the model with both strategic awareness from the macro router and fine-grained adaptivity from the micro router, leading to more precise and context-sensitive expert selection.

## 4.2 DYNAMIC MOE MOMENTUM UPDATE

In a continual VLN setting, an ideal agent must assimilate knowledge from the current task $t$ while preserving the general navigation capabilities acquired from previous tasks $1, \ldots, t-1$. To achieve this balance, we introduce a Dynamic MoE Momentum Update mechanism that differentiates experts by their contribution to task $t$ and updates their parameters with task-aware momentum.

**Step 1: Expert workload and contribution.** For each MoE layer, denote by $\mathbf{w}(x) \in \mathbb{R}^n$ the fused routing weights of a token $x \in D_t$ obtained from the dual-router fusion in Eq. equation 8. We first accumulate per-expert workload over all tokens of $D_t$,

$$\mathbf{u} = \sum_{x \in D_t} \mathbf{w}(x) \in \mathbb{R}^n, \tag{9}$$

and normalize it to obtain a contribution distribution,

$$\mathcal{I}_t(E_i) \triangleq \mathbf{p}[i] = \frac{\mathbf{u}[i]}{\sum_{j=1}^n \mathbf{u}[j]} = \frac{\sum_{x \in D_t} w_i(x)}{\sum_{j=1}^n \sum_{x \in D_t} w_j(x)} \tag{10}$$

Here, $\mathcal{I}_t(E_i)$ is the normalized importance of expert $E_i$ on task $t$.

**Step 2: Top-$K$ expert selection.** We select the $K$ most active experts as important experts and treat the rest as non-important:

$$\mathcal{E}_t^{\mathrm{imp}} = \mathrm{TopK}(\mathbf{p}, K), \qquad \mathcal{E}_t^{\mathrm{non}} = \{E_1, \ldots, E_n\} \setminus \mathcal{E}_t^{\mathrm{imp}}. \tag{11}$$

**Step 3: Momentum-based parameter consolidation.** Let $\Theta_{t-1}$ be the consolidated expert parameters from previous tasks and $\Phi_t$ the parameters obtained by fine-tuning on $D_t$ (initialized from $\Theta_{t-1}$). We assign a momentum coefficient $\lambda_i$ to each expert $E_i$ based on its group:

$$\lambda_i = \begin{cases} \gamma, & \text{if } E_i \in \mathcal{E}_t^{\mathrm{imp}}, \\ 1 - \gamma, & \text{if } E_i \in \mathcal{E}_t^{\mathrm{non}}, \end{cases} \qquad \gamma \in [0, 0.5), \tag{12}$$

so that important experts lean more toward $\Phi_t$ (aggressive adaptation), while non-important experts preserve $\Theta_{t-1}$ (conservative update). Let $\mathbf{\Lambda} = [\lambda_1, \ldots, \lambda_n]$ denote the per-expert momentum vector (broadcast to the parameter shapes). The consolidated parameters are then computed by an element-wise interpolation:

$$\Theta_t \;=\; \mathbf{\Lambda} \odot \Theta_{t-1} \;+\; (\mathbf{1} - \mathbf{\Lambda}) \odot \Phi_t, \tag{13}$$

where $\odot$ denotes element-wise multiplication and $\mathbf{1}$ is the all-ones mask.

This task-aware momentum consolidation adapts important experts rapidly to the current task while preserving generalizable knowledge in non-important experts, thereby mitigating catastrophic forgetting and improving transferability in sequential navigation.

## 5 EXPERIMENTS

**Datasets & Protocol.** We evaluate the proposed method on R2R Anderson et al. (2018) and REVERIE Qi et al. (2020) (goal-oriented, object-anchored instructions) in Matterport3D photo-realistic indoor environments. In R2R, instructions are received step-by-step, while REVERIE always receives goal-oriented and object-anchored instructions. For each dataset, we adopt a *scene-partitioned continual stream*. Concretely, a *base agent* is first pretrained on the `train` split. Its performance is then evaluated on the `val-seen` split, which consists of previously encountered environments but novel instructions. This evaluation, denoted as *BaseAgent0*, serves as a reference for in-distribution capability, and is further used to assess both performance on trained environments and generalization to new instructions within them.

We then partition the `val-unseen` scenes by `scene_id` into $D$ disjoint domains $\{\mathcal{S}_1, \ldots, \mathcal{S}_D\}$. Each domain defines one continual-learning task $\mathcal{T}_d$, where all episodes belong to environments in $\mathcal{S}_d$. To construct reliable per-task splits, we adopt an 8:2 ratio for training and validation within each `val-unseen` domain, while discarding scenes with fewer than or equal 10 validation episodes to ensure sufficient validation(see Appendix A.1).

At stage $d \in \{1, \ldots, D\}$, the agent is initialized with consolidated parameters $\Theta_{d-1}$ and fine-tuned only on the training data of $\mathcal{T}_d$, without accessing prior-task data (replay-free continual learning). After finishing $\mathcal{T}_d$, the updated policy is evaluated on all tasks $0{:}d$ to track both adaptation to new domains and retention of prior knowledge; after the final stage, we additionally report performance on all $D$ tasks. To control training budgets across methods, the number of optimization steps for each task is fixed, with the same early-stopping and validation protocol. Episode sampling follows standard VLN practice, with a fixed maximum step limit per episode and candidate viewpoint set. This protocol provides a controlled, privacy-friendly benchmark for continual learning in VLN, stressing cross-domain adaptation and catastrophic forgetting.

**Evaluation Metrics.** We report both standard VLN and continual-learning metrics.

VLN metrics: (i) *Success Rate (SR)* — the fraction of episodes where the agent stops within 3m of the goal; (ii) *Success weighted by Path Length (SPL)* — SR penalized by the ratio between the agent's trajectory length and the shortest path; (iii) *Navigation Error (NE)* — the average distance (meters) between the stopping point and the goal; (iv) *Oracle Success Rate (OSR)* — the fraction of episodes in which any visited viewpoint is within 3m of the goal.

Continual learning metrics: After completing training on task $t$, the policy is evaluated on all tasks $0{:}t$ using the above VLN metrics. Let $R_{t,i}$ denote the performance on task $i$ after training up to task $t$. We report the continual-learning metrics:

$$\mathrm{BWT} = \frac{1}{T-1}\sum_{i=0}^{T-1}\left(R_{T,i} - R_{i,i}\right), \mathrm{FWT} = \frac{1}{T-1}\sum_{i=2}^{T}\left(R_{i-1,i} - R_{i,i}\right). \tag{14}$$

Here, Backward Transfer (BWT) measures how learning new tasks influences performance on previous tasks, while Forward Transfer (FWT) evaluates zero-shot generalization by testing each task before it is trained. Note that, unlike Li et al. (2024), our definition of BWT explicitly includes the *BaseAgent* ($i = 0$), so that it also measures the extent of forgetting relative to the initial in-distribution capability.

**Baselines.** To demonstrate the effectiveness of our method, we compare against several representative continual learning strategies, built on the same LLM-based navigation backbone (Sec. 3.2),

Table 1: Domain-incremental learning in R2R environment. Methods are categorized by strategy: Reg (regularization), Reh (rehearsal), and RF (replay-free).

| Method | Strategy | | | AvgSR% ↑ | AvgSPL% ↑ | AvgNE ↓ | BWT ↑ | FWT ↑ |
|---|---|---|---|---|---|---|---|---|
| | Reg | Reh | RF | | | | | |
| Finetune | × | × | ✓ | 63.28 | 59.08 | 3.72 | -5.42 | -2.41 |
| L2 | ✓ | × | × | 58.78 | 56.20 | 4.23 | -5.10 | -3.43 |
| EWC | ✓ | × | × | 64.15 | 60.21 | 3.60 | -3.50 | -2.80 |
| ER | × | ✓ | × | 66.35 | 62.10 | 3.45 | -1.50 | 0.50 |
| PerR | × | ✓ | × | 67.05 | 62.93 | 3.38 | -1.35 | 0.62 |
| ESR | × | ✓ | × | 68.12 | 63.88 | 3.25 | -1.10 | 0.85 |
| Dual-SR | ✓ | ✓ | × | 70.25 | 65.40 | 3.05 | -0.45 | 1.85 |
| **M³E (ours)** | × | × | ✓ | **71.92** | **66.96** | **2.95** | **0.04** | **2.15** |

specifically NaviLLM Zheng et al. (2024). For regularization-based methods, we adopt L2 and EWC Kirkpatrick et al. (2017), which constrain parameter updates to preserve prior knowledge. For rehearsal-based methods, we evaluate ER Schaul et al. (2015), PerR, ESR Jeong et al. (2024), and Dual-SR Li et al. (2024), which mitigate forgetting by storing and replaying past trajectories with different sampling strategies. As a replay-free baseline, we include naive *Finetune*, which simply trains sequentially without any CL mechanism. Our proposed M³E belongs to the replay-free category, introducing a dual-router MoE and momentum-based consolidation to achieve continual adaptation without replay. Please see Table 1 for the categorization of methods.

**Implementation Details.** All methods are built on the NaviLLM backbone Zheng et al. (2024) with our MoE–LoRA integration. We use a total of 6 experts, with Top-$K = 2$ routing and fusion weight $\beta = 0.3$. For momentum-based consolidation, we set $\gamma = 0.4$ for task-critical experts (Sec. 4.2). LoRA adapters use rank $r = 16$ with scaling factor $\alpha = 32$, injected into both feed-forward (FFN) and attention projection layers. Training is performed with a batch size of 4 and a learning rate of $2 \times 10^{-5}$. Each task is trained with five epochs and AdamW optimization. For rehearsal-based baselines, the memory bank size is fixed to 100 episodes per task. All experiments are conducted on NVIDIA A800 80GB GPUs.

## 5.1 MAIN RESULTS

**Comparison with continual-learning baselines.** Table 1 reports averaged navigation performance across domains (AvgSR/AvgSPL/AvgNE) and continual learning metrics (BWT/FWT) on R2R. All methods share the same training budget, replay-based methods report results with their buffer memory included in the accounting.

Table 2: Domain-incremental learning in REVERIE environment.

| Method | SR% ↑ | SPL% ↑ | BWT ↑ | FWT ↑ |
|---|---|---|---|---|
| Finetune | 50.12 | 39.86 | -16.91 | -10.26 |
| **M³E (ours)** | **51.23** | **48.30** | **-5.91** | **-8.09** |

From the results, we highlight the following observations. First, our method M³E achieves the best overall performance among all strategies, outperforming both regularization-based and rehearsal-based baselines. For instance, it improves AvgSPL by $+1.56\%$ over the strongest rehearsal method Dual-SR (66.96 vs. 65.40), while also reducing AvgNE (2.95 vs. 3.05). Second, unlike most baselines that suffer from large negative BWT (e.g., -5.42 for Finetune, -3.50 for EWC), our approach achieves near-zero BWT (0.04), showing that it effectively mitigates catastrophic forgetting without replay. Third, M³E also demonstrates strong forward transfer (FWT = 2.15), indicating improved zero-shot generalization to new domains before training. Fourth, while rehearsal-based methods (ER, PerR, ESR, Dual-SR) alleviate forgetting to some extent, they rely on storing past trajectories, raising scalability and privacy concerns. In contrast, our replay-free method achieves better performance with no access to previous-task data. In addition to R2R, we also evaluate on the more challenging REVERIE dataset (Tab. 2). While other methods suffer from severe forgetting in this goal-oriented setting, M³E substantially alleviates degradation: compared to Finetune, it improves SPL by $+8.44\%$ (48.30 vs. 39.86) and reduces forgetting (BWT $-5.91$ vs. $-16.91$). This demonstrates that our replay-free design also transfers effectively to object-centric navigation tasks, where precise grounding and long-horizon reasoning are required. Overall, these findings show that M³E not only improves navigation success but also establishes a stronger balance between generalization and knowledge retention, setting a new state-of-the-art replay-free baseline.

**Forgetting and generalization under full `val-unseen` training.** Table 3 evaluates a different setting from our continual-learning protocol. Instead of task-by-task domain-incremental

Table 3: Performance before and after training on the full `val-unseen` split. *Before* denotes the released checkpoint; *After* denotes the performance after post training; $\Delta$ denotes the gain from post training (*After*−*Before*).

| Method | State | R2R | | | | REVERIE | | | |
| | | val-seen | | test-unseen | | val-seen | | test-unseen | |
| | | SR% ↑ | SPL% ↑ | SR% ↑ | SPL% ↑ | SR% ↑ | SPL% ↑ | SR% ↑ | SPL% ↑ |
|---|---|---|---|---|---|---|---|---|---|
| **HAMT** | *Before* | 70.13 | 66.30 | 62.30 | 57.57 | 43.29 | 40.19 | 33.41 | 26.67 |
| | *After* | 57.59 | 52.67 | 58.01 | 52.06 | 42.38 | 37.90 | 28.67 | 24.23 |
| | $\Delta$ | -12.54 | -13.63 | -4.29 | -5.51 | -0.91 | -2.29 | -4.74 | -2.44 |
| **NaviLLM** | *Before* | 73.36 | 68.93 | 67.60 | 59.84 | 50.11 | 47.24 | 39.53 | 32.51 |
| | *After* | 71.11 | 67.97 | 63.52 | 58.27 | 38.93 | 36.05 | 34.98 | 28.93 |
| | $\Delta$ | -2.25 | -0.96 | -4.08 | -1.57 | -11.18 | -11.19 | -4.55 | -3.58 |
| **M$^3$E (ours)** | *Before* | 73.36 | 68.93 | 67.60 | 59.84 | 50.11 | 47.24 | 39.53 | 32.51 |
| | *After* | **75.51** | **71.94** | **67.28** | **61.33** | **46.24** | **42.93** | **37.79** | **31.64** |
| | $\Delta$ | +2.15 | +3.01 | -0.32 | +1.49 | -3.87 | -4.31 | -1.74 | -0.87 |

training, we directly load the officially released checkpoints and continue training on the entire `val-unseen` split. This setup probes two aspects: (*i*) how much knowledge about previously seen environments (`val-seen`) is forgotten, and (*ii*) how the overall generalization to novel environments (`test-unseen`) changes after further training.

As shown in the table, HAMT suffers from substantial forgetting on `val-seen` (e.g., $-12.54$ SR on R2R) and clear drops on unseen scenes. NaviLLM exhibits milder degradation on R2R but severe forgetting on REVERIE, especially in val-seen performance ($-11.18$ SR). In contrast, our M$^3$E achieves positive gains on R2R val-seen and maintains competitive performance on test-unseen, while exhibiting much smaller drops on REVERIE compared to NaviLLM. These results suggest that M$^3$E not only resists catastrophic forgetting when adapting to new domains in bulk training but also preserves cross-domain generalization better than both strong baselines.

## 5.2 ABLATION STUDY

Table 4 reports a comprehensive ablation of the three core components on R2R, covering all eight possible combinations. Starting from naive finetuning, the agent suffers from catastrophic forgetting (BWT $-5.42$) and poor transfer. Applying the *Momentum Update* directly to the dense backbone (equivalent to EMA) improves retention (BWT $-2.15$) but restricts plasticity, lowering the success rate to $61.52\%$; this indicates that stabilization without specialized expert routing is insufficient for adapting to new domains. Introducing the *Micro Router* or *Macro Router* individually improves navigation accuracy (SR $65.51\%$) via token-level specialization and generalization (FWT $+1.80$) through topology-aware reasoning, respectively, yet forgetting remains severe without consolidation. Hybrid combinations reveal distinct trade-offs: *Macro+Momentum* secures structural knowledge (FWT $+1.92$) but lacks fine-grained grounding, while *Micro+Momentum* improves accuracy but limits transfer. Notably, the dual-router design (*Micro+Macro*) yields the strongest plasticity ($67.83\%$) among non-momentum methods but fails to retain past knowledge (BWT $-6.05$). Only when all three components are combined does the model achieve the best overall trade-off, reaching $71.92\%$ SR and nearly eliminating forgetting (BWT $\approx 0$). This confirms that routing (Micro+Macro) and momentum-based consolidation are complementary: the former improves specialization and generalization, and the latter ensures stability across domains.

## 6 CONCLUSION

We presented M$^3$E, a replay-free continual learning framework for Vision-and-Language Navigation that separates global scene reasoning from local instruction–vision grounding via a macro–micro Mixture-of-Experts. Built on a trainable LLM-based agent, M$^3$E uses dual routing for expert activation and consolidates task-critical experts through differentiated momentum, enabling fast adaptation without storing past data. Evaluated on R2R and REVERIE under a scene-partitioned protocol, M$^3$E improves navigation performance and shows stronger forward and backward transfer compared to rehearsal- and regularization-based baselines.

Table 4: Ablations on R2R under domain-incremental learning. We evaluate all combinations of the Micro Router, Macro Router, and Dynamic MoE Momentum Update. Note that applying Momentum without routers (Row 2) is equivalent to applying Exponential Moving Average (EMA) to the dense model.

| Components | | | R2R Metrics | | | | |
|---|---|---|---|---|---|---|---|
| Micro | Macro | Momentum | AvgSR (%) ↑ | AvgSPL (%) ↑ | AvgNE ↓ | BWT ↑ | FWT ↑ |
| × | × | × | 63.28 | 59.08 | 3.72 | -5.42 | -2.41 |
| × | × | ✓ | 61.52 | 57.90 | 3.95 | -2.15 | -2.80 |
| × | ✓ | × | 64.20 | 60.15 | 3.68 | -5.65 | 1.80 |
| × | ✓ | ✓ | 65.15 | 61.05 | 3.60 | -0.12 | 1.92 |
| ✓ | × | × | 65.51 | 61.23 | 3.55 | -5.81 | -1.50 |
| ✓ | × | ✓ | 66.72 | 62.34 | 3.48 | -0.35 | -1.48 |
| ✓ | ✓ | × | 67.83 | 64.12 | 3.34 | -6.05 | 2.15 |
| ✓ | ✓ | ✓ | **71.92** | **66.96** | **2.95** | **0.04** | **2.15** |

**Limitations and future work.** First, while our experiments are conducted in discrete simulators to adhere to standard VLN protocols, the proposed macro–micro decomposition is inherently compatible with continuous control. In such settings, $M^3E$ functions as a high-level waypoint planner for local controllers (such as ROS MoveBase). Bridging the Sim-to-Real gap requires further addressing challenges such as visual domain gaps (e.g., lighting, motion blur) and real-world SLAM estimation errors, though our robustness analysis (Appendix A.3) suggests resilience to map noise. Second, the fixed expert configuration may limit flexibility under extreme open-ended scenarios. Future work includes dynamic expert allocation, privacy-aware hybrid replay, Sim-to-Real transfer and extension to physical robotic platforms.

## ACKNOWLEDGMENTS

The work is supported by National Natural Science Foundation of China (No.62302170), Guangdong Basic and Applied Basic Research Foundation (No.2024A1515010187), Guangdong Natural Science Funds for Distinguished Young Scholars (Grant 2023B1515020097), the Singapore Ministry of Education Academic Research Fund Tier 2 (Award No. MOE-T2EP20125-0016), the Singapore Ministry of Education Academic Research Fund Tier 1 (Proposal ID: 24-SIS-SMU-015), and the Lee Kong Chian Fellowships.

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

# A APPENDIX

## A.1 INCREMENTAL DATASET CONSTRUCTION

We construct incremental datasets for both R2R and REVERIE following the same protocol. For clarity, we use R2R as an illustrative example below.

### A.1.1 R2R DATASET STATISTICS.

Table 5 summarizes the official R2R splits. The training set contains 61 scans with 15,060 instructions and 4,675 paths. The validation splits are divided into *val-seen* and *val-unseen*. *Val-seen* consists of 56 scans that overlap with training environments but have disjoint paths and instructions (1,021 instructions, 340 paths). *Val-unseen* contains 11 novel scans that are completely disjoint from both train and val-seen. The test split further includes 18 novel scans (1,391 paths, 4,173 instructions) that do not overlap with any other split.

Table 5: R2R dataset splits and statistics.

| Split | Scans | Paths | Instructions | Notes |
|---|---|---|---|---|
| Train | 61 | 4,675 | 15,060 | standard training set |
| Val-seen | 56 | 340 | 1,021 | overlapping scans, new paths/instructions |
| Val-unseen | 11 | 783 | 2,349 | disjoint scans from all other splits |
| Test-unseen | 18 | 1,391 | 4,173 | disjoint scans from all other splits |

**Val-unseen distribution.** Within *val-unseen*, the number of trajectories per scan is highly imbalanced. The top scans include 2azQ1b91cZZ, oLBMNvg9in8, TbHJrupSAjP, and zsNo4HB9uLZ (each with ∼100 paths), whereas smaller scans such as 8194nk5LbLH (15 paths) and pLe4wQe7qrG (6 paths) contain very few samples.

Table 6: Statistics of R2R val-unseen scans and their corresponding incremental dataset entries. Scans with ≤ 10 validation trajectories are excluded from the incremental benchmark (shown in gray).

| Scan ID | # Trajectories (R2R) | Train entries | Val entries |
|---|---|---|---|
| 2azQ1b91cZZ | 100 | 80 | 20 |
| oLBMNvg9in8 | 100 | 80 | 20 |
| TbHJrupSAjP | 100 | 80 | 20 |
| zsNo4HB9uLZ | 100 | 80 | 20 |
| X7HyMhZNoso | 98 | 78 | 20 |
| QUCTc6BB5sX | 93 | 74 | 19 |
| EU6Fwq7SyZv | 64 | 51 | 13 |
| Z6MFQCViBuw | 60 | 48 | 12 |
| x8F5xyUWy9e | 47 | 37 | 10 |
| 8194nk5LbLH | 15 | 12 | 3 |
| pLe4wQe7qrG | 6 | 4 | 2 |
| **Total** | **883** | **624** | **159** |

**Incremental split construction.** To construct the incremental benchmark, we sort val-unseen scans by trajectory count and adopt an 8:2 split at the scene level. Scans with less than or equal to 10 validation trajectories are removed to ensure reliable validation performance. The resulting incremental dataset provides: (i) a training stream for continual learning evaluation, and (ii) a held-out validation set that is sufficiently large and diverse to measure both VLN navigation accuracy and continual learning metrics (BWT/FWT).

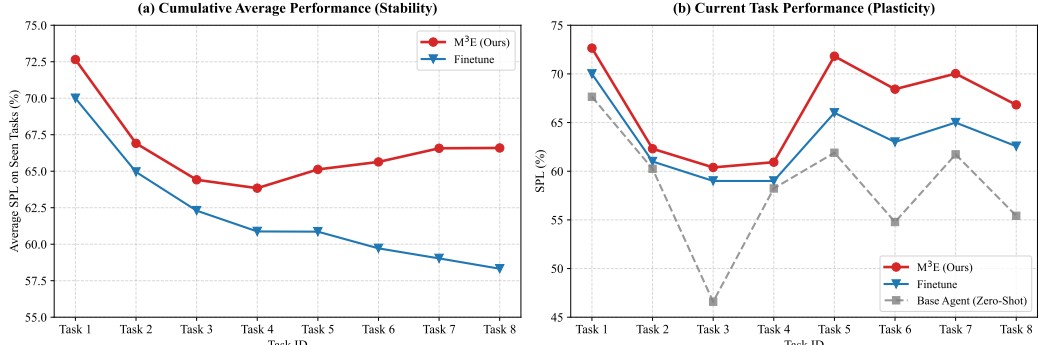

Figure 2: Performance Trend Analysis. (a) Cumulative Stability: The Finetune baseline shows a downward trend due to interference, whereas M³E maintains a stable upward trend, indicating effective knowledge accumulation. (b) Plasticity: M³E consistently achieves higher performance on the current task being learned compared to Finetune, demonstrating sustained learning capability.

## A.2 SCALABILITY AND PERFORMANCE TREND ANALYSIS

To further validate the effectiveness of M³E in continuous adaptation, we analyze the temporal dynamics of the learning process. Verifying performance trends across episodes is crucial for establishing the scalability of continual learning agents.

**Metric for Performance Trend.** To quantify the performance trend across lifelong learning stages, we adopt the *Cumulative Average SPL on Seen Tasks* ($A_t$) as the primary metric. Let $R_{t,i}$ denote the SPL on task $i$ after the agent has finished training on task $t$. The metric at stage $t$ is defined as:

$$A_t = \frac{1}{t+1} \sum_{i=0}^{t} R_{t,i} \tag{15}$$

This metric dynamically tracks how well the agent performs on the entirety of domains encountered so far. A non-decreasing $A_t$ curve indicates that the agent successfully learns new tasks (plasticity) without degrading on previous ones (stability).

**Empirical Evidence of Scalability.** We plot the trend of $A_t$ in Figure 2 and the detailed performance matrix in Figure 3.

*Performance Trend (Figure 2):* In terms of Cumulative Stability (Left), the Finetune baseline (blue line) exhibits a monotonic downward trend in $A_t$, dropping from ∼70% to ∼59% due to severe interference. In contrast, M³E (red line) stabilizes after the initial tasks and exhibits a gradual upward trend from Task 4 to Task 8. This confirms that our method effectively accumulates knowledge and scales gracefully as it experiences more episodes. Regarding Plasticity (Right), M³E consistently maintains high performance on the current task (red line > blue line), demonstrating that the model's ability to learn new, complex scenes does not degrade over time due to capacity saturation.

*Lifelong Transfer Matrix (Figure 3):* The heatmaps visualize $R_{t,i}$ across all 8 sequential tasks. For Finetune (Right), the lower-triangular region (representing past tasks) fades rapidly (color turns lighter) as training progresses, visually confirming catastrophic forgetting. For M³E (Left), the color intensity remains consistent and deep along the columns (e.g., Task 1 performance remains high even at Stage 8). Furthermore, the diagonal elements (learning new tasks) remain high throughout the curriculum. This empirically proves that the expert-isolation mechanism prevents capacity saturation, ensuring late-stage learning is as effective as early-stage learning.

This analysis confirms that M³E scales effectively. The decoupled expert architecture allows the agent to expand its domain coverage continuously, maintaining high performance on both early and late tasks.

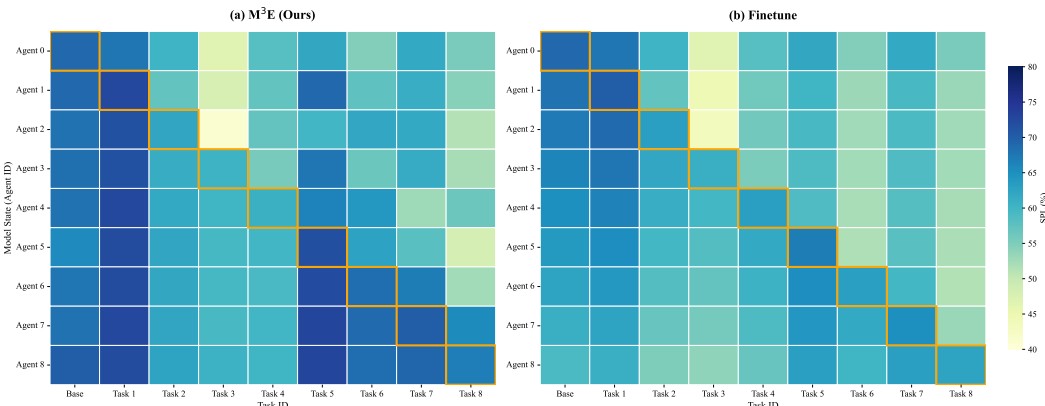

Figure 3: Lifelong Transfer Matrix ($R_{t,i}$) visualization. Each cell $(t, i)$ represents the SPL on Task $i$ after training on Task $t$. **Left (M³E):** The color intensity remains consistent and deep along the columns, indicating robust retention of past tasks. **Right (Finetune):** The lower-triangular region fades rapidly, visually confirming catastrophic forgetting.

## A.3 ROBUSTNESS ANALYSIS: RESILIENCE TO MAP IMPERFECTIONS

To address concerns regarding the dependency on accurate cognitive mapping, we empirically analyze the system's resilience to map imperfections. We demonstrate robustness through both architectural design and stress-test experiments on the R2R dataset.

**Architectural Resilience.** The M³E framework does not rely solely on the cognitive map. The final routing weight is a fusion: $\mathbf{w} = \beta \mathbf{w}^{ma} + (1 - \beta) \mathbf{w}^{mi}$. In our implementation, we set $\beta = 0.3$, meaning the routing decision is heavily weighted (70%) towards the **Micro Router**, which operates on immediate visual observations and instruction tokens. This design ensures graceful degradation: even if the cognitive map is noisy or incomplete (impairing $\mathbf{w}^{ma}$), the Micro Router continues to guide the agent based on local perception, preventing catastrophic failure.

**Empirical Evidence.** We conducted a robustness analysis by introducing varying levels of noise to the cognitive map input of the Macro Router during inference:

- **Edge Drop**: Randomly removing 10% to 30% of edges from the topology graph to simulate connectivity errors.

- **No Frontiers**: Removing all frontier nodes, simulating poor exploration awareness.

As shown in Table 7, even under severe noise conditions such as a 30% edge drop, the performance degradation is minimal (∼1.1 absolute points in SR). Furthermore, the robust M³E continues to significantly outperform the Micro-only baseline (70.83% vs. 66.72%), confirming that even a degraded cognitive map provides valuable structural context compared to having no map at all.

Table 7: Robustness of M³E under Cognitive Map Noise (R2R). Performance drops are minimal compared to the complete removal of the Macro Router (Micro-only).

| Setting | Noise Type | SR (%) ↑ | SPL (%) ↑ | △ SR (Points) |
|---|---|---|---|---|
| **Full M³E (Standard)** | None (Original) | **71.92** | **66.96** | - |
| M³E (Robustness) | Edge Drop (10%) | 71.65 | 66.58 | -0.27 |
| M³E (Robustness) | Edge Drop (30%) | 70.83 | 65.90 | -1.09 |
| M³E (Robustness) | No Frontier Nodes | 70.42 | 65.21 | -1.50 |
| **Micro-only Baseline** | Macro Router Disabled | 66.72 | 62.34 | -5.20 |

## A.4 ABLATION ON EXPERT CAPACITY

To verify whether the fixed expert configuration limits performance under domain shifts, we conducted an ablation study on the R2R dataset with varying numbers of experts ($N \in \{4, 6, 8\}$).

As shown in Table 8, increasing the expert count from 4 to 6 yields a significant gain (+1.77% SR), indicating that sufficient capacity is needed to capture diverse navigation primitives. However, further increasing from 6 to 8 results in diminishing returns (+0.13% SR). This saturation indicates that the capacity of 6 experts is sufficient to cover the domain variations in standard benchmarks. Furthermore, while the total parameter count increases with more experts, our Top-K routing ($K = 2$) ensures that the **inference computational cost remains constant**. Thus, our chosen configuration of 6 experts represents an optimal trade-off between performance and efficiency.

Table 8: Impact of Expert Count on R2R Performance. Increasing experts beyond 6 yields diminishing returns.

| # Experts | Trainable Params | Avg SR (%) ↑ | Avg SPL (%) ↑ | BWT (%) ↑ | FWT (%) ↑ |
|---|---|---|---|---|---|
| 4 | 226.0M | 70.15 | 65.10 | -1.12 | 1.85 |
| **6 (Ours)** | **310.6M** | **71.92** | **66.96** | **0.04** | **2.15** |
| 8 | 395.1M | 72.05 | 67.08 | 0.05 | 2.18 |

## A.5 RELATION TO TEST-TIME ADAPTATION (TTA)

To explicitly position M³E against recent Test-Time Adaptation (TTA) approaches in VLN, such as FSTTA (ICML '24)Gao et al. (2024) and FeedTTA (ICML '25)Kim et al., we clarify the theoretical distinctions and provide a quantitative comparison.

**Theoretical Positioning.** We distinguish M³E from TTA methods along three dimensions:

- **Task Setting (Lifelong vs. Transient):** TTA methods focus on inference-time adaptation to a single incoming stream. M³E targets lifelong learning across distinct domains, where retaining knowledge of past domains (BWT) is critical.

- **Supervision Dependency:** FeedTTA relies on an external Oracle (success/failure feedback) during testing. M³E is self-contained and requires no external signals during inference.

- **Mechanism:** TTA methods often use gradient regularization to smooth updates on shared parameters. M³E uses architectural decoupling (MoE) to structurally isolate knowledge, providing robust long-term memory.

**Quantitative Comparison.** We adapted FSTTA and FeedTTA to our R2R benchmark under a **Continual TTA (CTTA)** protocol using LoRA. As shown in Table 9, while TTA methods improve adaptation (AvgSPL) over naive finetuning, they suffer from severe catastrophic forgetting (negative BWT) because they destructively update shared parameters. In contrast, M³E achieves the highest performance while uniquely maintaining positive BWT.

Table 9: Comparison with TTA methods on R2R Continual Stream. TTA methods suffer from forgetting (negative BWT) in lifelong settings.

| Method | Type | Test-time Input | AvgSPL (%) ↑ | BWT ↑ |
|---|---|---|---|---|
| **Finetune** (Baseline) | CL | Frozen | 59.08 | -5.42 |
| **FSTTA** (ICML'24) | TTA | Unlabeled Stream | 63.45 | -6.04 |
| **FeedTTA** (ICML'25) | TTA | Stream + Oracle Feedback | 63.84 | -5.78 |
| **M³E (Ours)** | **CL** | **Frozen** | **66.96** | **0.04** |

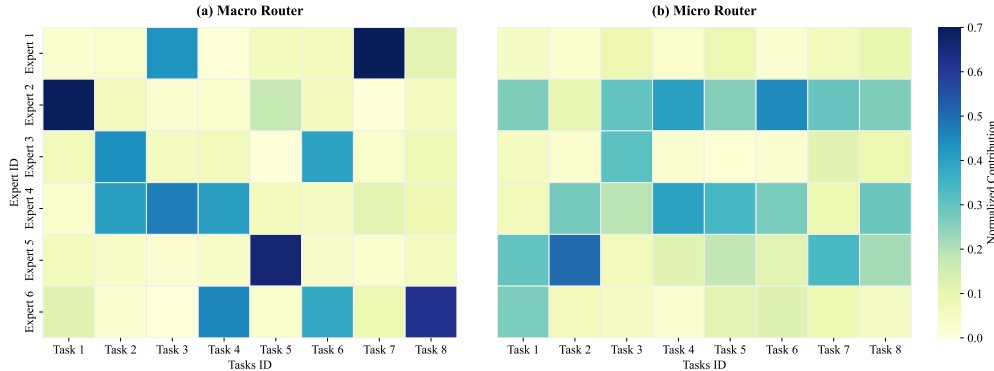

Figure 4: Visualization of Expert Specialization across 8 sequential domains. **(a) Macro Router** exhibits domain-specific activation shifts, indicating it captures changing scene contexts. **(b) Micro Router** shows consistent activation across tasks, indicating it captures shared linguistic and visual semantics.

## A.6 VISUALIZATION OF MACRO-MICRO DISENTANGLEMENT

To empirically verify our dual-router design and demonstrate the effective disentanglement of global (scene) and local (instruction) reasoning, we analyzed the Expert Contribution Distribution (as defined in Eq. 10) across 8 distinct domains in our domain-incremental training stream.

**Visualization of Expert Specialization.** We visualized the activation heatmaps for both routers across the 8 sequential tasks. As shown in Figure 4, the results reveal two distinct patterns that validate our architectural hypothesis:

- **Macro Router: Task-Specific Sparsity (Figure 4a).** The Macro experts exhibit distinct shifts in activation as the agent moves between domains. For example, Expert 2 dominates in Task 1 but drops significantly in Task 2, where Expert 3 and Expert 4 take over. Similarly, Expert 5 is highly specialized for Task 5. This sparsity and variability confirm that the Macro Router effectively captures *Global Scene Context*—such as specific topological layouts or room types unique to each environment—and routes to experts specialized for those structural priors.

- **Micro Router: Cross-Task Consistency (Figure 4b).** In sharp contrast, the Micro experts display a consistent activation pattern across all 8 domains. For instance, Expert 2, Expert 4, and Expert 5 maintain high activity levels throughout the entire stream (visible as horizontal bands). This aligns with our design that the Micro Router focuses on *Local Semantics* (e.g., grounding verbs like "go" or nouns like "door"), which are linguistic and visual primitives that generalize across different environments.

Overall, this contrast—Macro experts shifting with the domain versus Micro experts remaining stable—provides strong empirical evidence that $M^3E$ successfully disentangles scene-level structural reasoning from token-level perceptual grounding.

Table 10: Inference Latency Breakdown per Step (ms).

| Map Size ($N$) | Macro Router (GNN) | Micro Router (MLP) | LLM Backbone (7B) | Relative Overhead |
|---|---|---|---|---|
| 10 nodes | 0.80 ms | 0.17 ms | 152.10 ms | **+0.5%** |
| 50 nodes | 3.07 ms | 0.17 ms | 152.48 ms | **+2.0%** |
| 100 nodes | 6.10 ms | 0.17 ms | 152.80 ms | **+4.0%** |

## A.7 INFERENCE LATENCY PROFILING

To quantify the exact overhead, we profiled the inference latency of different components on a single NVIDIA A800 GPU. We measured the per-step processing time as the cognitive map size

($N$) increases from 10 to 100. As shown in Table 10, the Macro Router introduces approximately 3 ms of overhead for a typical map size ($N = 50$), which is orders of magnitude smaller than the LLM decoding time ($\sim$152 ms). The relative overhead is only around 2%, which does not create a computational bottleneck.

## LLM USAGE STATEMENT

Large Language Models (LLMs) were employed to aid in the writing and polishing of this manuscript. Specifically, we used an LLM to refine the language, improve readability, and ensure clarity in selected sections. The model's assistance included tasks such as sentence rephrasing, grammar checking, and enhancing the overall flow of the text.

It is important to emphasize that the LLM was **not** involved in ideation, research methodology, experimental design, or analysis. All research concepts, ideas, and results presented in this paper were entirely developed and conducted by the authors. The LLM's role was limited to improving the linguistic presentation of the paper, without any contribution to the scientific content or data interpretation.

The authors take full responsibility for the entirety of the manuscript, including any text generated or refined with the help of the LLM. We have ensured that the use of the LLM complies with ethical standards, does not involve plagiarism, and does not constitute scientific misconduct.

