# OpenReview forum: "M$^3$E: Continual Vision-and-Language Navigation via Mixture of Macro and Micro Experts"
_ICLR.cc/2026/Conference — ICLR 2026 Poster_

### Official Review · Reviewer_tcXs · 2025-10-27

**Soundness:** 3
**Presentation:** 3
**Contribution:** 2
**Rating:** 4
**Confidence:** 4

**Summary:**

This paper introduces a new framework for continual learning in Vision-and-Language Navigation (VLN) tasks. The core problem addressed is catastrophic forgetting, where an agent's performance on previously learned environments degrades after being trained on new ones. The proposed solution is a replay-free method based on a Mixture-of-Experts (MoE) architecture integrated into an LLM-based navigation agent.

The key innovation is a dual-router system that separates reasoning into two streams: a Macro Router for global, scene-level strategic planning using a graph neural network over a cognitive map, and a Micro Router for local, token-level semantic grounding. To consolidate knowledge without replaying old data, the authors propose a Dynamic MoE Momentum Update mechanism that selectively updates or freezes expert parameters based on their utility in the current task.

The method is evaluated on a domain-incremental benchmark constructed from the R2R and REVERIE datasets, where it is shown to outperform standard fine-tuning and several continual learning baselines in both navigation success and knowledge retention.

**Strengths:**

To begin with, the paper tackles  a critical and well-recognized challenge that must be overcome to build autonomous systems capable of lifelong learning in dynamic environments. Decoupling high-level spatial reasoning from low-level instruction grounding is a very sensible approach to modularizing knowledge and seems well-suited to the hierarchical nature of navigation. And the  method's ability to mitigate forgetting without storing and replaying past trajectory data is a significant strength.

**Weaknesses:**

I have a few concerns that temper my enthusiasm.

If I understand well, the primary motivation of introducing continual learning to  VLN is explicitly tied to real-world agents that must continually adapt to new domains. However, the evaluation is performed exclusively in a simulated environment on an artificially constructed stream of tasks (partitioning the val-unseen set). I think this is a significant limitation, as the challenges of a simulated, sequential dataset may not fully represent the complexities of real-world continual learning. The paper would be much stronger (and from my point, it ought to contain it) if it discussed the potential challenges of Sim2Real transfer.

Another concern is that I wonder about the computational complexity of the Macro Router. It appears to construct a graph from the agent's cognitive map and perform GNN propagation at each decision step. As the agent explores a large environment, this map could grow significantly. More discussion on the computational overhead and scalability of this component seems necessary for a complete picture of the method's practicality.

**Questions:**

(1) Could the authors comment on the challenges they foresee in deploying your method on a physical robot?

(2) Could the authors provide some analysis of the inference-time overhead introduced by the Macro Router, especially as the size of the cognitive map increases throughout an episode?

---

> ### Author Response · Authors · 2025-11-26
> **Response to Reviewer tcXs (1/2)**
>
> > **W4.1**: If I understand well, the primary motivation of introducing continual learning to VLN is explicitly tied to real-world agents that must continually adapt to new domains. However, the evaluation is performed exclusively in a simulated environment on an artificially constructed stream of tasks (partitioning the val-unseen set). I think this is a significant limitation, as the challenges of a simulated, sequential dataset may not fully represent the complexities of real-world continual learning. The paper would be much stronger (and from my point, it ought to contain it) if it discussed the potential challenges of Sim2Real transfer.
> >
> > **Q4.1**: Could the authors comment on the challenges they foresee in deploying your method on a physical robot?
>
> **AW4.1 & AQ4.1:** We thank the reviewer for this insightful comment. We fully agree that real-world deployment introduces complexities beyond simulation. While our evaluation focuses on algorithmic benchmarking of continual learning in a controlled environment, we have carefully considered the path to physical deployment.
>
> We utilize VLN simulated benchmarks as a standard and fair testbed to isolate the problem of catastrophic forgetting without the confound of hardware noise. However, our domain-incremental stream is constructed by partitioning buildings to mimic a realistic deployment pattern where a robot is introduced to new environments over time.
>
> The design of $M^3E$ is intentionally aligned with real robotic systems. The Macro Router consumes topological maps compatible with standard Graph-SLAM pipelines, while the hierarchical decoupling mirrors the standard global planner plus local controller architecture. Regarding specific deployment challenges, we identify three key areas: mapping noise, perceptual gaps, and continuous control.
>
> First, real-world SLAM is prone to estimation errors. To address this, we conducted a robustness analysis by injecting noise into the cognitive map such as randomly dropping 30% of edges. As shown in Table R5, $M^3E$ maintains stable performance with a success rate drop of only 1.09 points, confirming resilience to drift. We have included this robustness analysis in Appendix A.3. Second, regarding visual gaps, the modular architecture of $M^3E$ allows swapping visual encoders for robust models to handle lighting variations. Third, for continuous control, $M^3E$ functions as a high-level planner outputting subgoals, while standard local controllers such as ROS MoveBase handle precise motion. We have added a discussion on these Sim2Real challenges in the Limitations section.
>
> **Table R5: Robustness of $M^3E$ under Cognitive Map Noise (R2R)**
>
> | **Setting**                | **Noise Type**        | **SR (%) ↑** | **SPL (%) ↑** | **Δ SR (Points)** |
> | -------------------------- | --------------------- | ------------ | ------------- | :---------------: |
> | **Full $M^3E$ (Standard)** | None (Original)       | **71.92**    | **66.96**     |         -         |
> | **$M^3E$ (Robustness)**    | Edge Drop (10%)       | 71.65        | 66.58         |       -0.27       |
> | **$M^3E$ (Robustness)**    | Edge Drop (30%)       | 70.83        | 65.90         |       -1.09       |
> | **$M^3E$ (Robustness)**    | No Frontier Nodes     | 70.42        | 65.21         |       -1.50       |
> | **Micro-only Baseline**    | Macro Router Disabled | 66.72        | 62.34         |       -5.20       |

---

> ### Author Response · Authors · 2025-11-26
> **Response to Reviewer tcXs (2/2)**
>
> > **W4.2**: Another concern is that I wonder about the computational complexity of the Macro Router. It appears to construct a graph from the agent's cognitive map and perform GNN propagation at each decision step. As the agent explores a large environment, this map could grow significantly. More discussion on the computational overhead and scalability of this component seems necessary for a complete picture of the method's practicality.
> >
> > **Q4.2**: Could the authors provide some analysis of the inference-time overhead introduced by the Macro Router, especially as the size of the cognitive map increases throughout an episode?
>
> **AW4.2 & AQ4.2:** We appreciate the reviewer's attention to the computational practicality of our method. We acknowledge that the Macro Router adds overhead, but both analysis and profiling confirm it is negligible. Theoretically, the cognitive map size in standard VLN episodes rarely exceeds 50 nodes, making the GNN operations computationally lightweight compared to the substantial cost of the 7B LLM backbone.
>
> To quantify this, we profiled the inference latency on an NVIDIA A800 GPU as the map size increases. As shown in Table R6, the Macro Router introduces approximately 3 ms of overhead for a typical map size of 50 nodes, which is orders of magnitude smaller than the LLM decoding time of roughly 152 ms. Even with 100 nodes, the relative overhead is only around 4%, which does not create a computational bottleneck. We have included this latency analysis in Appendix A.7.
>
> **Table R6: Inference Latency Breakdown per Step (ms)**
>
> | **Map Size (N)** | **Macro Router (GNN)** | **Micro Router (MLP)** | **LLM Backbone (7B)** | **Relative Overhead** |
> | :--------------- | :--------------------- | :--------------------- | :-------------------- | :-------------------- |
> | 10 nodes         | 0.80 ms                | 0.17 ms                | 152.10 ms             | +0.5%                 |
> | 50 nodes         | 3.07 ms                | 0.17 ms                | 152.48 ms             | +2.0%                 |
> | 100 nodes        | 6.10 ms                | 0.17 ms                | 152.80 ms             | +4.0%                 |
>
> ---
>
> **To Reviewer tcXs: We believe that the concerns raised regarding real-world deployment challenges and computational complexity have been addressed in this rebuttal through our new Sim2Real robustness analysis and inference latency profiling. We sincerely request a fair re-evaluation of our submission in light of these clarifications and additional results.**

---

### Official Review · Reviewer_Jsee · 2025-11-01

**Soundness:** 3
**Presentation:** 3
**Contribution:** 2
**Rating:** 4
**Confidence:** 3

**Summary:**

This paper presents M3E, a replay-free continual learning framework for Vision-and-Language Navigation (VLN).
M3E is designed to disentangle global scene reasoning from local instruction–vision grounding via a macro–micro Mixture-of-Experts (MoE) structure.
The framework further introduces a dual routing mechanism for expert activation and a differentiated momentum update to consolidate task-critical experts without relying on replay data.
Experiments conducted on R2R and REVERIE demonstrate that M3E achieves improved navigation performance and alleviates catastrophic forgetting compared to existing continual learning approaches.

**Strengths:**

* The paper provides a clear motivation for leveraging a macro–micro MoE strategy to enhance transferability and stability in lifelong VLN tasks.

* The proposed topology-aware propagation using GNNs and the dual-routing fusion methods are conceptually simple yet appears to be an effective way to capture structural relationships in navigation environments.

* The replay-free continual learning design aligns well with realistic online settings, making the proposed approach practically meaningful.

**Weaknesses:**

* The paper lacks a review and comparison with recent online test-time adaptation approaches for VLN, such as FeedTTA (ICML '25) and FSTTA (ICML '24). In particular, FeedTTA also investigates catastrophic forgetting through a gradient regularization mechanism, which is conceptually similar to the replay-free and momentum-based update proposed in this paper. Please consider positioning M3E more explicitly in relation to these methods to highlight its distinct contributions by providing quantitative comparisons if possible.

* The topology-aware, task-focused dual routing mechanism is interesting, but the individual contributions of each component remain unclear. For example, Tab.3 does not isolate the effects of the macro-level router or the momentum update in ablation studies, making it difficult to assess their independent impact.

* While the dual-router fusion mechanism is intuitively reasonable, it remains unclear how effectively the model disentangles and learns global versus local information in practice. The authors should provide analytical or visualization-based evidence (e.g., expert activation maps or attention distributions per task/environment changes) to support the claim that such separation emerges through the proposed design.

At this stage, I lean toward a borderline rejection. However, I'm willing to increase the score based upon the authors responses and further clarifications.

**Questions:**

Please refer to the weakness section where I also listed questions and suggestions.

---

> ### Author Response · Authors · 2025-11-26
> **Response to Reviewer Jsee (1/3)**
>
> > **W3.1**: The paper lacks a review and comparison with recent online test-time adaptation approaches for VLN, such as FeedTTA (ICML '25) and FSTTA (ICML '24). In particular, FeedTTA also investigates catastrophic forgetting through a gradient regularization mechanism, which is conceptually similar to the replay-free and momentum-based update proposed in this paper. Please consider positioning M3E more explicitly in relation to these methods to highlight its distinct contributions by providing quantitative comparisons if possible.
>
> **AW3.1:** We thank the reviewer for pointing out these relevant recent works. We distinguish $M^3E$ from TTA methods such as FeedTTA and FSTTA along three dimensions: task setting, supervision dependency, and mechanism. First, TTA methods focus on inference-time adaptation to a single incoming stream, whereas $M^3E$ targets lifelong learning across distinct domains where retaining knowledge of past domains is critical. Second, approaches such as FeedTTA rely on external signals or oracle feedback during testing, while $M^3E$ is self-contained. Third, and most importantly regarding your comment on regularization, TTA methods use gradient regularization as a soft constraint on shared parameters. In contrast, $M^3E$ uses architectural decoupling to structurally isolate knowledge, providing a more robust mechanism for long-term memory than gradient heuristics.
>
> To empirically highlight these differences, we adapted FSTTA and FeedTTA to our R2R benchmark under a Continual TTA protocol using LoRA to ensure a fair comparison. As shown in Table R3, while TTA methods improve adaptation over Finetune, they suffer from severe catastrophic forgetting because they destructively update shared parameters. In contrast, $M^3E$ achieves the highest overall performance while uniquely maintaining positive BWT. We have included this quantitative comparison in Appendix A.5 to explicitly position $M^3E$ against these approaches.
>
> **Table R3: Comparison with TTA methods on R2R Continual Stream**
>
> | **Method**              | **Type** | **Test-time Input**      | **AvgSPL (%) ↑** | **BWT ↑** |
> | :---------------------- | :------- | :----------------------- | :--------------- | :-------- |
> | **Finetune** (Baseline) | CL       | Frozen                   | 59.08            | -5.42     |
> | **FSTTA** (ICML'24)     | TTA      | Unlabeled Stream         | 63.45            | -6.04     |
> | **FeedTTA** (ICML'25)   | TTA      | Stream + Oracle Feedback | 63.84            | -5.78     |
> | **$M^3E$ (Ours)**       | **CL**   | **Frozen**               | **66.96**        | **0.04**  |

---

> ### Author Response · Authors · 2025-11-26
> **Response to Reviewer Jsee (2/3)**
>
> > **W3.2**: The topology-aware, task-focused dual routing mechanism is interesting, but the individual contributions of each component remain unclear. For example, Tab.3 does not isolate the effects of the macro-level router or the momentum update in ablation studies, making it difficult to assess their independent impact.
>
> **AW3.2:** We thank the reviewer for highlighting the need to isolate the contributions of each component. We previously omitted the isolated Momentum-only ablation because our Dynamic MoE Momentum relies on expert contribution scores derived from the routers. To address this, we conducted a comprehensive 8-row ablation study covering all component combinations. Specifically, for the Momentum-only setting in Row 2, we applied standard Exponential Moving Average (EMA) to the dense backbone to simulate stability consolidation without expert specialization.
>
> The new results reveal clear, independent contributions of each component. First, applying Momentum or EMA alone improves retention but restricts plasticity, lowering the success rate to 61.52%. This confirms that stabilization without expert routing is insufficient for adapting to new domains. Second, deploying the routers individually highlights their distinct roles. The Macro Router significantly boosts zero-shot generalization via FWT through topology awareness, while the Micro Router improves instruction following accuracy via precise token grounding. However, without momentum consolidation, both configurations suffer from severe catastrophic forgetting. Finally, only the full framework achieves the best overall trade-off, reaching 71.92% SR and near-zero forgetting. This synergy proves that the dual routers provide the necessary specialization and generalization, while the momentum update ensures stability across domains. We have updated the ablation study in Section 5.2 and Table 4 to include these detailed comparisons.
>
> **Table R4: Ablations on R2R under domain-incremental learning**
>
> | Micro | Macro | Momentum | AvgSR (%) ↑ | AvgSPL (%) ↑ | AvgNE ↓  |  BWT ↑   |  FWT ↑   |
> | :---: | :---: | :------: | :---------: | :----------: | :------: | :------: | :------: |
> |   -   |   -   |    -     |    63.28    |    59.08     |   3.72   |  -5.42   |  -2.41   |
> |   -   |   -   |    ✓     |    61.52    |    57.90     |   3.95   |  -2.15   |  -2.80   |
> |   -   |   ✓   |    -     |    64.20    |    60.15     |   3.68   |  -5.65   |   1.80   |
> |   -   |   ✓   |    ✓     |    65.15    |    61.05     |   3.60   |  -0.12   |   1.92   |
> |   ✓   |   -   |    -     |    65.51    |    61.23     |   3.55   |  -5.81   |  -1.50   |
> |   ✓   |   -   |    ✓     |    66.72    |    62.34     |   3.48   |  -0.35   |  -1.48   |
> |   ✓   |   ✓   |    -     |    67.83    |    64.12     |   3.34   |  -6.05   |   2.15   |
> | **✓** | **✓** |  **✓**   |  **71.92**  |  **66.96**   | **2.95** | **0.04** | **2.15** |
>
> ------
>
> > **W3.3**: While the dual-router fusion mechanism is intuitively reasonable, it remains unclear how effectively the model disentangles and learns global versus local information in practice. The authors should provide analytical or visualization-based evidence (e.g., expert activation maps or attention distributions per task/environment changes) to support the claim that such separation emerges through the proposed design.
>
> **AW3.3:** We thank the reviewer for raising this critical point regarding the empirical verification of our dual-router design. To demonstrate the effective disentanglement of global scene and local instruction reasoning, we analyzed the expert contribution distribution across 8 distinct domains in our training stream.
>
> We visualized the activation heatmaps for both routers across the 8 sequential tasks as shown in Figure 4 in Appendix A.6. The results reveal two distinct patterns that validate our architectural hypothesis. First, the Macro experts exhibit distinct shifts in activation as the agent moves between domains. For example, Expert 2 dominates in Task 1 but drops significantly in Task 2 where Expert 3 takes over. This sparsity confirms that the Macro Router captures global scene context such as specific topological layouts unique to each environment.
>
> In sharp contrast, the Micro experts display a consistent activation pattern across all 8 domains. Experts 2, 4, and 5 maintain high activity levels throughout the entire stream. This aligns with our design that the Micro Router focuses on local semantics such as grounding verbs or nouns, which are primitives that generalize across different environments. Overall, this contrast provides strong empirical evidence that $M^3E$ successfully disentangles scene-level structural reasoning from token-level perceptual grounding. We added these visualization heatmaps and the corresponding analysis to Appendix A.6.

---

> ### Author Response · Authors · 2025-11-26
> **Response to Reviewer Jsee (3/3)**
>
> **To Reviewer Jsee: We appreciate the constructive feedback regarding comparisons with test-time adaptation methods, the isolation of component contributions, and the verification of our dual-router design. We have addressed these by adding quantitative TTA baselines, a comprehensive ablation study, and expert visualization analysis in the revision. We sincerely request a fair re-evaluation of our submission in light of these additional results.**

---

> > ### Comment · Reviewer_Jsee · 2025-11-27
> >
> > I found the authors rebuttal addresses most of my original concerns and increased the rating from 4 to 6.
> > Please incorporate the the additional experiments and explanation into the paper.

---

### Official Review · Reviewer_eMkh · 2025-11-01

**Soundness:** 3
**Presentation:** 3
**Contribution:** 2
**Rating:** 6
**Confidence:** 3

**Summary:**

In this paper, the authors propose M3E (Mixture of Macro and Micro Experts) for continual end-to-end vision-language navigation. For replay-free continual learning, the authors use dynamic momentum update for the MoE layers by selecting the top-performing experts, which helps them prevent catastrophic forgetting. The navigation policy consists of a visual scene encoder and a transformer decoder as a policy, with the FFNs replaced with M3E. The architecture is dual router: There is a global router captures global structural regularities of the environment and uses a GNN to build spatial relationships between different nodes in the scene graph and eventually pass it to experts; the local router performs token-wise routing over N experts.

They experiment with R2R and REVERIE benchmark in MP3D datasets. They evaluate in both seen and unseen environments. In addition to traditional VLN metrics (SR, SPL, NE, OSR), they also compute continual learning metrics (backward transfer, forward transfer). The baselines used are fine-tuned policies, L2, EWC, ER, etc, consisting of regularization-based and rehearsal-based methods. M3E performs the best, achieving the highest average SPL, reducing average navigation error significantly, and having lesser back-ward transfer error showing that it prevents catastrophic forgetting. They also perform additional experiments to train on val-unseen and show that the performance on val-seen does not reduce, while maintaining competitive performance on test-unseen, against HAMT and NaviLLM. They also perform ablations on micro/macro routers and momentum based updates and show that all of them are important for best performance.

**Strengths:**

1. The authors tackle a relatively under-explored problem of continual learning and catastrophic forgetting in vision and language navigation, and propose a methodical approach to solve it.
2. The idea of decoupling scene level and token level information is pretty important and useful for handling the continual learning task.
3. The approach actually helps in preventing catastrophic forgetting across different environments as shown by low BWT metric.

**Weaknesses:**

1. The $M^{3}E$ framework "relies on accurate cognitive mapping". This implies that the performance of the macro-router, and thus the entire system, could degrade if the underlying cognitive map representation is poor, especially since it relies on a topological graph.
2. The method uses fixed expert configurations. This may limit flexibility under extreme domain shifts, as the agent cannot dynamically add new experts if a new environment is radically different from all previous ones.
3. The authors do not discuss how the approach is applicable to settings beyond VLN - VLN is a hard problem, but there are existing approaches which perform very well on these tasks.
4. While BWT is near-zero on R2R, the method is not a perfect for preventing forgetting. In the "full val-unseen training" experiment (Table 3), $M^{3}E$ still shows some performance degradation (e.g., -4.31 SPL, -3.87 SR) on the REVERIE val-seen split, though it is much more resilient than the NaviLLM baseline.

**Questions:**

1. How easy is it to apply this approach to other tasks such as mobile-manipulation or table-top manipulation?

---

> ### Author Response · Authors · 2025-11-26
> **Response to Reviewer eMkh (1/2)**
>
> > **W2.1**: The framework relies on accurate cognitive mapping. This implies that the performance of the macro-router, and thus the entire system, could degrade if the underlying cognitive map representation is poor, especially since it relies on a topological graph.
>
> **AW2.1:** We appreciate the reviewer’s concern about map quality. While a better cognitive map helps the Macro Router, $M^3E$ is designed to degrade gracefully when the map is noisy. First, the final routing is a fusion $\mathbf{w} = \beta \mathbf{w}^{ma} + (1-\beta)\mathbf{w}^{mi}$, and we use $\beta=0.3$, so routing is primarily driven by the Micro Router, which depends only on the current observation and instruction tokens.
>
> Second, to empirically verify this resilience, we stress-tested map imperfections on R2R by perturbing the macro graph during inference as detailed in Appendix A.3. We introduced noise types such as random edge drops and removing frontier nodes. As shown in Table R1, even with a 30% edge drop, the SR decrease is small, and performance remains clearly higher than disabling the Macro Router entirely, indicating that the macro signal remains useful even when degraded.
>
> **Table R1: Robustness of $M^3E$ under Cognitive Map Noise (R2R)**
>
> | **Setting**            | **Noise Type**        | **SR (%) ↑** | **SPL (%) ↑** | **Δ SR (Points)** |
> | ---------------------- | --------------------- | ------------ | ------------- | ----------------- |
> | Full $M^3E$ (Standard) | None (Original)       | **71.92**    | **66.96**     | -                 |
> | $M^3E$ (Robustness)    | Edge Drop (10%)       | 71.65        | 66.58         | -0.27             |
> | $M^3E$ (Robustness)    | Edge Drop (30%)       | 70.83        | 65.90         | -1.09             |
> | $M^3E$ (Robustness)    | No Frontier Nodes     | 70.42        | 65.21         | -1.50             |
> | Micro-only Baseline    | Macro Router Disabled | 66.72        | 62.34         | -5.20             |
>
> ------
>
> > **W2.2**: The method uses fixed expert configurations. This may limit flexibility under extreme domain shifts, as the agent cannot dynamically add new experts if a new environment is radically different from all previous ones.
>
> **AW2.2**: We appreciate the reviewer's comment on expert configuration. While dynamic expansion is a promising direction, we deliberately chose a fixed configuration to balance flexibility with inference efficiency. Theoretically, experts learn reusable navigation primitives such as obstacle avoidance rather than memorizing environments, allowing the model to handle diverse domains through unique weightings of shared primitives. Practically, this ensures constant inference cost due to Top-K routing, avoiding the unbounded parameter growth seen in dynamic methods.
>
> To verify if this limits performance, we conducted an ablation study on the R2R dataset with varying expert counts as detailed in Appendix A.4. As shown in Table R2, increasing the expert count from 4 to 6 yields a significant gain (+1.77% SR). However, increasing from 6 to 8 results in diminishing returns (+0.13% SR). This saturation indicates that 6 experts are sufficient to cover domain variations in standard benchmarks. We have included this ablation study in Appendix A.4 to justify our design choice.
>
> **Table R2: Impact of Expert Count on R2R Performance**
>
> | **Number of Experts** | **Trainable Params (M)** | **Avg SR (%) ↑** | **Avg SPL (%) ↑** | **BWT (%) ↑** | **FWT (%) ↑** |
> | :--- | :--- | :--- | :--- | :--- | :--- |
> | 4 | 226.0M | 70.15 | 65.10 | -1.12 | 1.85 |
> | **6 (Ours)** | **310.6M** | **71.92** | **66.96** | **0.04** | **2.15** |
> | 8 | 395.1M | 72.05 | 67.08 | 0.05 | 2.18 |

---

> ### Author Response · Authors · 2025-11-26
> **Response to Reviewer eMkh (2/2)**
>
> > **W2.3**: The authors do not discuss how the approach is applicable to settings beyond VLN — VLN is a hard problem, but there are existing approaches which perform very well on these tasks.
> >
> > **Q2.1**: How easy is it to apply this approach to other tasks such as mobile-manipulation or table-top manipulation?
>
> **AW2.3 & Q2.1:** We thank the reviewer for this insightful suggestion. We address the applicability to broader embodied domains together with the question on manipulation tasks.
>
> The core philosophy of $M^3E$, decoupling macro-level strategic reasoning from micro-level perceptual grounding, is a universal principle in robotics and is highly applicable beyond VLN. Applying $M^3E$ to tasks such as mobile or table-top manipulation is architecturally straightforward because most modern policies such as RT-2 share Transformer backbones.
>
> For mobile manipulation, the Macro Router can take a semantic map or scene graph as input to route experts for high-level skills, while the Micro Router processes proprioception to handle end-effector control. Similarly, for table-top manipulation, the Macro Router can focus on task progress states to specialize in distinct temporal stages such as grasping versus lifting, while the Micro Router manages object-specific physical interactions. The implementation effectively functions as a plug-and-play module where only the domain-specific macro input needs definition.
>
> ---
>
> > **W2.4**: While BWT is near-zero on R2R, the method is not perfect for preventing forgetting. In the "full val-unseen training" experiment (Table 3), M3E still shows some performance degradation (e.g., -4.31 SPL, -3.87 SR) on the REVERIE val-seen split, though it is much more resilient than the NaviLLM baseline.
>
> **AW2.4:** We appreciate the reviewer's fair assessment. We acknowledge that while $M^3E$ achieves near-zero forgetting on structure-dominant tasks such as R2R, it exhibits residual degradation on the more challenging REVERIE dataset. This degradation stems from the inherent difficulty of visual object grounding under extreme domain shifts. Unlike structural features in R2R, REVERIE requires recognizing specific objects whose appearances vary drastically across domains, such as chair styles. This forces the Micro experts to adapt aggressively to learn new visual patterns, incurring a slightly higher cost to stability compared to structural features.
>
> However, it is crucial to contextualize this against the difficulty of the replay-free setting. As shown in Table 3, the strong baseline NaviLLM suffers a catastrophic drop of 11.18% in success rate, whereas $M^3E$ limits this drop to only 3.87%. This corresponds to a 65% reduction in forgetting. While not yet perfect, our method shields the vast majority of task-critical knowledge compared to state-of-the-art baselines.
>
> ---
>
> **To Reviewer eMkh: We believe that the concerns raised regarding map dependency, expert flexibility, and generalization have been thoroughly addressed in this rebuttal through our new robustness experiments and ablation studies. We sincerely request a fair re-evaluation of our submission in light of these clarifications and additional results.**

---

### Official Review · Reviewer_P3Q9 · 2025-11-03

**Soundness:** 3
**Presentation:** 3
**Contribution:** 2
**Rating:** 4
**Confidence:** 4

**Summary:**

This paper presents a novel continual learning benchmark on vision - language navigation task and proposes a replay - free method - Mixture of Macro and Micro Experts (M3E) approach that effectively deals with the catastrophic forgetting problem in the continual learning setting. Two core components are responsible for the superior performance of the M3E approach: A macro - router conditioned on the scene context and a micro - router conditioned on egocentric observation. By designing a mixture - of - expert frameworks with both routers, the M3E is shown to work better than most of the continual learning methods.

**Strengths:**

1. The continual learning settings designed for the VLN task are not only theoretically interesting but also serve as a data-efficient strategy to enhance the generalization ability of VLN approaches.
2. The proposed method (M3E) is well-justified in addressing the catastrophic forgetting problem, particularly in the aforementioned continual learning settings for VLN.

**Weaknesses:**

1. The proposed method (M3E) relies on meta-information (i.e., scene topology) to construct the macro-router and adheres to conventional discrete VLN settings—both factors widen the gap for its practical real-robot deployment. Discrete settings, for instance, often fail to align with the continuous perceptual and motion requirements of physical robots, while the dependence on pre-defined scene topology limits adaptability to unstructured environments.
2. No evidence is provided in this paper to demonstrate that the proposed method can scale effectively under continual learning settings. Specifically, there is a lack of analysis on whether the model’s performance on episodes in later (or more complex) scenes remains comparable to, or outperforms, those in the first few scenes—an essential indicator of scalability in continual learning.
3. As shown in Table 4, the M3E fails to address the catastrophic forgetting problem under the REVERIE benchmark. This limitation makes it challenging to verify the model’s generalization ability across different VLN tasks, especially within the continual learning framework emphasized earlier.

**Questions:**

(1) Although the experimental results shown in Table 4 demonstrate that the M3E method can effectively solve (or greatly alleviate) the catastrophic forgetting problem, there is a lack of results to confirm whether the task performance of M3E continues to improve as it experiences more episodes under lifelong learning settings—consistent with the paper’s continual learning framework. Therefore, I wonder if there is a specific metric that quantifies the performance trend of M3E across different numbers of experienced episodes?
(2) In Table 4, the performance metrics of M3E on the R2R and REVERIE benchmarks are inconsistent (e.g., M3E alleviates forgetting on R2R but still suffers from it on REVERIE). What are the key reasons for this inconsistency in performance metrics between the two benchmarks? Furthermore, what primary factors cause M3E to still struggle with catastrophic forgetting in the REVERIE task?
(3) Currently, this work is built on the NaviLLM framework. Given this, can the M3E approach be generalized to other navigation foundation models? If not, what technical constraints limit its generalization across different foundation models?

---

> ### Author Response · Authors · 2025-11-26
> **Response to Reviewer P3Q9 (1/3)**
>
> ### Weaknesses
> > **W1.1**: The proposed method (M3E) relies on meta-information (i.e., scene topology) to construct the macro-router and adheres to conventional discrete VLN settings—both factors widen the gap for its practical real-robot deployment. Discrete settings, for instance, often fail to align with the continuous perceptual and motion requirements of physical robots, while the dependence on pre-defined scene topology limits adaptability to unstructured environments.
>
> **AW1.1**: We appreciate the reviewer's concern regarding real-world deployment. We clarify that our method does not depend on pre-defined scene topology or privileged meta-information. The cognitive map used by the Macro Router is constructed entirely online during navigation and comprises only visited nodes and observable frontier candidates. This structure aligns directly with standard real-world robotic representations such as Graph-SLAM and frontier-based exploration pipelines, which bridges rather than widens the sim-to-real gap.
>
> Regarding the discrete setting, we adhere to standard R2R and REVERIE protocols for fair comparison with baselines. However, our hierarchical design naturally generalizes to continuous spaces where $M^3E$ can function as high-level waypoint planner while a local controller such as ROS MoveBase handles continuous actuation.
>
> We have revised Section 4.1.1 to explicitly emphasize the online construction of the map and added a discussion on continuous control extensions in the Limitations section.
>
> ---
>
> > **W1.2**: No evidence is provided in this paper to demonstrate that the proposed method can scale effectively under continual learning settings. Specifically, there is a lack of analysis on whether the model’s performance on episodes in later (or more complex) scenes remains comparable to, or outperforms, those in the first few scenes—an essential indicator of scalability in continual learning.
> >
> > **Q1.1**: Although the experimental results shown in Table 4 demonstrate that the M3E method can effectively solve (or greatly alleviate) the catastrophic forgetting problem, there is a lack of results to confirm whether the task performance of M3E continues to improve as it experiences more episodes under lifelong learning settings. Is there a specific metric that quantifies the performance trend of M3E across different numbers of experienced episodes?
>
> **AW1.2 & AQ1.1**: We appreciate the reviewer's constructive suggestion to analyze the temporal dynamics and scalability of the learning process. To quantify the performance trend as requested in Q1.1, we adopt the Cumulative Average SPL on Seen Tasks ($A_t$) as the primary metric. This metric tracks how well the agent performs on the entirety of domains encountered so far.
>
> We have added a trend analysis in Appendix A.2 to demonstrate scalability. As shown in Figure 2, the Finetune baseline shows a monotonic downward trend in $A_t$, dropping from roughly 70% to 59% due to severe interference. In contrast, $M^3E$ stabilizes after the initial tasks and exhibits a gradual upward trend from Task 4 to Task 8. This confirms that our method effectively accumulates knowledge and scales gracefully as it experiences more episodes.
>
> Furthermore, the Lifelong Transfer Matrix in Figure 3 visually confirms this behavior. While the baseline's past task performance fades rapidly, $M^3E$ maintains consistent performance on earlier tasks such as Task 1 even at Stage 8. The diagonal elements also remain high throughout the curriculum, proving that the expert-isolation mechanism prevents capacity saturation and ensures late-stage learning remains effective.

---

> ### Author Response · Authors · 2025-11-26
> **Response to Reviewer P3Q9 (2/3)**
>
> > **W3**: As shown in Table 4, the M3E fails to address the catastrophic forgetting problem under the REVERIE benchmark. This limitation makes it challenging to verify the model’s generalization ability across different VLN tasks, especially within the continual learning framework emphasized earlier.
> >
> > **Q2**: In Table 4, the performance metrics of M3E on the R2R and REVERIE benchmarks are inconsistent (e.g., M3E alleviates forgetting on R2R but still suffers from it on REVERIE). What are the key reasons for this inconsistency? What factors cause M3E to still struggle with catastrophic forgetting in REVERIE?
>
> **AW1.3 & AQ1.2**: We appreciate the reviewer's careful examination of the results. Regarding the performance on REVERIE, we clarify that our method significantly mitigates forgetting rather than failing to address it. As shown in the experimental results, the standard Finetune method suffers from severe catastrophic forgetting with a BWT of -16.91%, while $M^3E$ reduces this to -5.91%. This corresponds to a 65% reduction in forgetting without storing or replaying any past data, which represents a substantial advancement in a strict replay-free setting.
>
> The difference in BWT between R2R and REVERIE from the fundamental difference between structure-dominant and object-dominant tasks. R2R focuses on movement through structural layouts such as hallways and stairs, which share high visual and topological similarity across domains. Our Macro Router effectively captures these stable patterns to maintain performance. In contrast, REVERIE requires grounding specific objects. The visual appearance of objects varies drastically across domains, such as a chair in an office versus one in a bedroom. To recognize these novel object variations, the Micro experts must undergo aggressive adaptation. Without data replay to anchor the visual features of old objects, this necessary plasticity leads to slightly higher forgetting compared to the stable structural reasoning in R2R.
>
> ------
>
> ### Questions
>
> > **Q3**: This work is built on the NaviLLM framework. Can the M3E approach be generalized to other navigation foundation models? If not, what technical constraints limit its generalization?
>
> **AQ1.3:** We thank the reviewer for this insightful question. We clarify that $M^3E$ is not restricted to NaviLLM and its design is intentionally model-agnostic. The core components, specifically the MoE-LoRA layers and the dual routing mechanism, are generic architectural modifications applicable to standard Transformer blocks. Most contemporary navigation foundation models such as NaVid and NavCoT share the same backbone pattern of a multimodal encoder followed by a Transformer decoder. $M^3E$ can be incorporated into these models by replacing their feed-forward layers with our MoE layers. The Micro Router operates on hidden states intrinsic to any Transformer, while the Macro Router relies on standard cognitive maps available in most embodied agents. Thus, there are no specific technical constraints limiting its generalization.

---

> ### Author Response · Authors · 2025-11-26
> **Response to Reviewer P3Q9 (3/3)**
>
> **To Reviewer P3Q9: We believe that the concerns raised regarding real-world deployment and scalability have been thoroughly addressed in this rebuttal through our clarifications on online mapping and the new trend analysis. We sincerely request a fair re-evaluation of our submission in light of these clarifications and additional results.**

---

> > ### Comment · Reviewer_P3Q9 · 2025-11-28
> >
> > Thank you for the authors' response, which addresses most of my concerns. Accordingly, I will raise my score.

---

### Author Response · Authors · 2025-11-26
**Response to All Reviewers**

We sincerely thank all reviewers for their thorough and constructive feedback. We are encouraged that the reviewers found our core idea of decoupling macro-micro reasoning to be "novel and interesting" (P3Q9), "pretty important and useful" (eMkh), and a "very sensible approach" (tcXs) to the significant problem of catastrophic forgetting. We also appreciate the recognition of our method as "well-justified" (P3Q9), "methodical" (eMkh), and "practically meaningful" (Jsee), with metrics confirming it "effectively deals with the catastrophic forgetting problem" (P3Q9).

The primary concerns raised across the reviews relate to (1) real-world deployment challenges (Sim2Real transfer and map dependency), (2) scalability and long-term performance trends, and (3) the need for deeper analysis of component contributions and comparisons with related TTA methods. Importantly, these critiques point towards the need for more comprehensive empirical validation rather than fundamental flaws in the proposed architecture. We have addressed each point with new experiments, detailed analyses, and clarifications, highlighting all modifications in the revised paper in blue.

We believe that these additional results and clarifications have resolved the reviewers' concerns regarding the practicality, scalability, and distinctiveness of our method. We respectfully request a re-evaluation of our submission in light of these improvements.

---

### Meta-Review · Area_Chair_HqmU · 2026-01-05

**Summary:**

This paper presents a hierarchical MoE framework for continual VLN, decoupling macro and micro reasoning to mitigate catastrophic forgetting.

Initial reviews raised concerns regarding real-world applicability (map dependency, Sim2Real), scalability, computational overhead, and insufficient ablation/analysis. The authors' rebuttal addressed these: they clarified the cognitive map is built online, demonstrated minor performance degradation under mapping noise, and provided latency analysis showing minimal overhead. New experiments include detailed ablations confirming component contributions, comparisons with TTA methods, and visualizations validating the macro-micro separation. These efforts resolved key concerns.

A remaining consideration is the method's residual forgetting on the REVERIE benchmark, acknowledged by the authors as a challenge in replay-free settings.

**Reviewer Concerns:**

See above.

**Reviewer Scores:**

While some reviewers may raise their scores, others may not, as some issues, e.g., real-world deployment, may not be fully resolved.

---

### Decision · Program_Chairs · 2026-01-26

Accept (Poster)